# Tumor Size and Oncological Outcomes in Patients with Early Cervical Cancer Treated by Fertility Preservation Surgery: A Multicenter Retrospective Cohort Study

**DOI:** 10.3390/cancers14092108

**Published:** 2022-04-23

**Authors:** Blanca Gil-Ibañez, Antonio Gil-Moreno, Aureli Torné, Angel Martín Jimenez, Mikel Gorostidi, Ignacio Zapardiel, Alvaro Tejerizo Garcia, Berta Diaz-Feijoo

**Affiliations:** 1Gynecological Oncology and Minimally Invasive Surgery Unit, Department of Obstetrics and Gynecology, 12 de Octubre University Hospital, 28041 Madrid, Spain; alvaro.tejerizo@salud.madrid.org; 2Gynecological Oncology Department, Hospital Universitari Vall d’Hebron, Universitat Autònoma de Barcelona, 08035 Barcelona, Spain; agil@vhebron.net; 3Centro de Investigación Biomédica en Red de Cáncer, CIBERONC, 28041 Madrid, Spain; 4Gynecologic Oncology Unit, Clinic Institute of Gynecology, Obstetrics, and Neonatology, Hospital Clinic of Barcelona, Institut d’Investigacions Biomèdiques August Pi i Sunyer (IDIBAPS), Universitat de Barcelona, 08036 Barcelona, Spain; atorne@clinic.cat (A.T.); bdiazfe@clinic.cat (B.D.-F.); 5Hospital Universitario Son Llàtzer, 07198 Mallorca, Spain; amartin@hsll.es; 6Obstetrics and Gynecology, Hospital Universitario de Donostia, 20014 San Sebastian, Spain; Miguelm.gorostidipulgar@osakidetza.eus; 7Gynecologic Oncology Unit, La Paz University Hospital, 28046 Madrid, Spain; ignaciozapardiel@alumni.harvard.edu

**Keywords:** early cervical cancer, fertility-sparing surgery, tumor size

## Abstract

**Simple Summary:**

As cervical cancer is increasingly diagnosed in women who still intend to have children, fertility-sparing surgery is arising as a treatment option for those women with early-stage cervical cancer. The aim of this study was to analyze surgical, oncological and obstetrical outcomes of fertility-sparing surgery in early cervical cancer in Spain. In our study, the tumor size was the most important negative prognostic factor in fertility-sparing surgery (FSS) in cervical cancer. Selection criteria for fertility preservation should be rigorous, especially for patients with a tumor > 2 cm, due to the worse oncological outcomes associated with such tumors. Patients with an early cervical cancer tumor > 2 cm and a desire for pregnancy should be advised against primary FSS.

**Abstract:**

Background: The aim of this study was to analyze the impact of tumor size > 2 cm on oncological outcomes of fertility-sparing surgery (FSS) in early cervical cancer in a Spanish cohort. Methods: A multicenter, retrospective cohort study of early cervical cancer (stage IA1 with lymphovascular space invasion -IB1 (FIGO 2009)) patients with gestational desire who underwent FSS at 12 tertiary departments of gynecology oncology between 01/2005 and 01/2019 throughout Spain. Results: A total of 111 patients were included, 82 (73.9%) with tumors < 2 cm and 29 (26.1%) with tumors 2–4 cm. Patients’ characteristics were balanced except from lymphovascular space invasion. All were intraoperative lymph node-negative. Median follow-up was 55.7 and 30.7 months, respectively. Eleven recurrences were diagnosed (9.9%), five (6.0%) and six (21.4%) (*p* < 0.05). The 3-year progression-free survival (PFS) was 95.7% (95%CI 87.3–98.6) and 76.9% (95% CI 55.2–89.0) (*p* = 0.011). Only tumor size (<2 cm vs. 2–4 cm) was found to be significant for recurrence. After adjusting for the rest of the variables, tumor size 2–4 cm showed a Hazard Ratio of 5.99 (CI 95% 1.01–35.41, *p* = 0.036). Conclusions: Tumor size ≥ 2 cm is the most important negative prognostic factor in this multicenter cohort of patients with early cervical cancer and gestational desire who underwent FSS in Spain.

## 1. Introduction

Approximately 15% of all cervical cancers occur in women under 40 years of age. More than half of these women are diagnosed with early-stage (FIGO 2009 stage IA1-IB1) tumors [1]. Traditionally, the treatment for early-stage cervical cancer has been radical hysterectomy with bilateral pelvic lymphadenectomy. In the current European guidelines, the level of radicality of the surgery depends on characteristics, such as tumor size, which play an important role in prognosis [2,3]. Radical hysterectomy has been shown to achieve excellent results, with a 5-year survival rate above 90% for these group of women, but with the cost of definitive infertility [4]. As cervical cancer is increasingly diagnosed in women who still intend to have children, fertility-sparing procedures, such as conization, trachelectomy or radical trachelectomy, are arising as treatments of choice for those women with early-stage cervical cancer.

In 1994, Dargent introduced the vaginal radical trachelectomy. Dargent’s procedure involved the resection of the cervix, the upper part of the vagina and the proximal part of the parametria via the vagina, combined with laparoscopic pelvic lymphadenectomy, whilst preserving the uterine corpus [5]. Dargent’s work has led to a new era of fertility-sparing surgery giving the option for women with early-stage cervical cancer to maintain their reproductive capacity without decreasing overall and recurrence-free survival [6,7]. These fertility-sparing surgeries have been mostly validated for tumors ≤ 2 cm. European cervical cancer guides [2] include fertility-sparing surgery exclusively for women with tumor size ≤ 2 cm, but the National Comprehensive Cancer Network’s cervical cancer guidelines [3] include radical trachelectomy as an option for selected young women who wish to preserve fertility and who have tumors up to 4 cm. However, none of these guidelines recommend a specific access route. In published studies, the selection criteria and approach for fertility-sparing surgery vary [8,9], and those focusing on outcomes of trachelectomy in tumors larger than 2 cm are few.

In Spain, due to better screening protocols and access to adequate treatment, cervical cancer 5-year net survival has considerably increased in the last two decades [10,11]. At the same time, the average age at maternity in Spain rises every year and was recorded in 2019 as 32.25 years old [12]. The combination of these two factors increases the trend of early cervical cancer diagnosis in women without a fulfilled gestational wish, making fertility-sparing surgery a necessary treatment option for young women.

The aim of this study was to analyze the impact of tumor size on surgical, oncological and obstetrical outcomes of fertility-sparing surgery in early cervical cancer in Spain.

## 2. Materials and Methods

A multicenter, retrospective cohort study of early cervical cancer patients with gestational desire who underwent fertility-sparing surgery was carried out at the departments of gynecological oncology of 12 referral cancer center hospitals between 01/2005 and 01/2019 throughout Spain. The inclusion criteria were as follows: (1) histologically confirmed squamous carcinoma, adenocarcinoma or adenosquamous carcinoma of cervical cancer; (2) stage IA1 with lymphovascular space invasion, IA2 and IB1 (<2 cm and 2–4 cm) (FIGO 2009); (3) patient’s fertility desire; and (4) no evidence of lymph node involvement and/or other metastasis in the preoperative imaging evaluation. The exclusion criteria were as follows: (1) other histologic subtypes; and (2) intraoperative frozen section analysis with confirmed positive lymph nodes or extrauterine disease.

A comparison of clinical data, histopathological findings, type of treatment and oncological and obstetrical outcomes was made between a study group of patients undergoing fertility-sparing surgery with tumor less than 2 cm and patients with tumor between 2–4 cm in greatest dimension.

Tumor size was evaluated based on the pathologic findings of the final specimen. In case of previous conization, the tumor size was evaluated by adding the measurements obtained in the conization and those of the residual disease in the trachelectomy specimen. The horizontal extent was measured by combining the maximum horizontal measurement in the different specimens and the depth of invasion measured as the maximum depth in either specimen.

Fertility-sparing surgery included simple trachelectomy and radical trachelectomy. The surgical approach (vaginal, laparotomy, laparoscopy or robotic assisted laparoscopy), level of radicality and the decision of whether to include sentinel lymph node mapping were based on surgeon and center protocols along the study time. Patients with tumor in the upper margin of a trachelectomy were assessed intraoperatively and underwent immediate new margin resection or radical hysterectomy. Cerclage placement was performed intraoperatively at the surgeon’s discretion.

Surgical outcomes were measured based on surgical reports. Intraoperative complications were recorded. Postoperative complications were considered during the first month, following the Clavien-Dindo classification [13]. Use of adjuvant chemotherapy or radiation were at the discretion of the gynecological oncology center, most of which followed the Sedlis criteria [14]. Progression-free survival was defined as time from surgery to diagnosis of local recurrence or metastasis. Overall survival was defined as time from surgery to date of death or last follow-up. Local recurrence was considered when lesions were located in the adnexal, in the cervical tissue and vaginal or paracervical areas, and lymph node recurrence was considered when positive nodes were located in the pelvic or aortic regions. Intraparenchymal metastasis was considered as distant recurrence. Obstetrical Outcomes were analyzed taking into account the global number of patients included in the study.

The data were registered in an online database where investigators included their cases. The study was approved by the Clinical Research Ethics Committee of Hospital Clinic (study protocol 87/2019) as the reference center and by the Institutional Review Boards of the participating hospitals. The study was carried out in accordance with the Declaration of Helsinki (7th revision) and the principles of good clinical practice.

Categorical variables are expressed as frequencies and percentages, and quantitative variables as mean and standard deviations. Differences in the distributions of clinicopathological characteristics among groups were analyzed by Student’s t-test for quantitative variables and with Chi squared test or Fisher’s exact test for qualitative variables as needed. Saphiro–Wilk and Kolgomorov–Smirnov tests were applied to check if the criteria of normality are met, and Levene test was used to check the homogeneity of variances. Progression-free and overall survival curves were estimated using the Kaplan–Meier method and were compared among groups using the log-rank test. Independent prognostic factors were assessed using univariate Cox proportional hazards regression analyses. Hazard ratios and their 95% confidence intervals are presented to quantify association. A two-sided *p* value of 0.05 was considered statistically significant. All analyses were performed using STATA 15 (StataCorp. 2017. Stata Statistical Software: Release 15. College Station, TX, USA: StataCorp LLC.) statistical software.

## 3. Results

A total of 111 patients who underwent trachelectomy were included, 82 (73.9%) of them with tumors <2 cm (were 5 patients with stage IA1 with lymphovascular space invasion and 15 patients with IA2) and 29 (26.1%) of them 2–4 cm. The basal characteristics of both populations are described in Table 1.

Patients who underwent trachelectomy with tumor <2 cm were on average older than patients with tumor 2–4 cm (33.7 vs. 30.0 years, *p* < 0.05). Other patients’ characteristics, except lymphovascular space invasion, were comparable between the two cohorts (Table 1).

### 3.1. SURGICAL Results

All patients were intraoperative node negative and therefore underwent cervical surgery. In one patient with a tumor 2–4 cm a micrometastasis was diagnosed at the time of ultrastaging. She underwent hysterectomy and received adjuvant therapy. There were no significant differences between groups regarding surgical approach, performance of a posterior cerclage, intraoperative complications or performance of posterior hysterectomy. No postoperative complications were recorded. A total of 24.1% of patients received adjuvant treatment in the group of tumors 2–4 cm because of risk factors (Table 1).

Median follow-up was 55.7 months in the patient group with ≤2 cm tumor and 30.7 months in the group with tumor 2–4 cm. Eleven recurrences were diagnosed at the time of the last follow up (9.9%), five (6.0%) in the <2 cm tumor group and six (21.4%) in the 2–4 cm tumor group. For these patients that had a recurrence in the group of tumor 2–4 cm, one of them underwent surgery with vaginal approach, four with laparoscopic approach and one robotic assisted-laparoscopic approach.

### 3.2. Oncological Results

The 3-year progression-free survival was 95.7% (95%CI 87.3–98.6) in patients with tumor < 2 cm and 76.9% (95% CI 55.2–89.0) in those with tumor 2–4 cm (log Rank: *p* = 0.011, Figure 1). Tumor size was divided into three categories (less than 20 mm vs. 20–29 mm vs. 30–40 mm), and analysis showed differences regarding PFS remained between tumors smaller or larger than 2 cm (log Rank: *p* = 0.026, Figure 2).

No differences were observed in terms of site of recurrence between tumors < 2 cm and tumors 2–4 cm (*p* = 0.84, Table 2). Significant differences in progression-free survival were also seen when excluding cervical recurrences from the analysis (three recurrences (3.75%) vs. five recurrences (17.86%) (*p* = 0.006).

Cox regression was performed to identify different predictor factor for recurrence. Only tumor size (< 2 cm vs. 2–4 cm) was found to be significant among histology, lymphovascular space invasion, previous conization or surgical approach. After adjusting for the rest of the variables, tumor size ≥ 2 cm has a Hazard Ratio of 5.99 (CI 95% 1.01–35.41, *p* = 0.036) Table 3.

At the time of the last follow up, 72.7% (8/11) of patients who recurred were free of disease, two patients were alive with disease and one patient died because of disease. This case occurred in a 25-year-old woman with a stage IB1 ≥ 2 cm (30 mm), lymphovascular space invasion-positive, squamous cell carcinoma of the cervix. She had a previous conization and underwent a robotic radical abdominal trachelectomy with cerclage after intraoperative negative lymphadenectomy. After a definitive pathology report with near margin infiltration, hysterectomy was offered to the patient, but she refused. At a 16-month follow-up, she presented a local recurrence that infiltrated the bladder. Chemoradiation was administrated without complete response and exenteration surgery was performed. Sixteen months later she died from her disease with carcinomatosis and distant lung metastasis.

### 3.3. Obstetrics Results

Regarding obstetric results, 29 of 111 patients got pregnant, with a total of 44 pregnancies. Thirty-one pregnancies were recorded in the < 2 cm group and thirteen pregnancies in the 2–4 cm group. There were sixteen first trimester abortions (38.7% vs. 30.8%) and three second trimester abortions (all in group of patients with tumor size 2–4 cm). There was a total of 13 preterm births (35.5% vs. 15.3%) and 12 full-term births (25.8% vs. 30.8%). All differences were not statistically significant.

## 4. Discussion

This study found that patients with early cervical cancer with tumor size ≥ 2 cm who underwent fertility-sparing surgery in Spain presented increased risk of recurrence, without differences in surgical nor in obstetrical outcomes, compared with those patients with tumor size < 2 cm with fertility-sparing surgery.

Despite the rising trend of delaying pregnancy in the early thirties in Spain, fertility preservation surgery is not very often offered after early cervical cancer diagnosis [15]. Although fertility-sparing surgery in cervical cancer should exclusively be undertaken in specific gynecologic-oncological centers with highly trained surgeons, every young woman with a desire to preserve her fertility should be counseled about the possibility of fertility-sparing surgery and be referred to a tertiary care hospital with comprehensive expertise in this kind of oncological therapy [2]. Nevertheless, the oncological outcomes of Spanish women with cervical cancer and tumor size < 2 cm are consistent and in concordance with those published in the literature [16]. In our study, the only independent factor that impacted the oncological outcomes after fertility-sparing surgery was tumor size. Patients who underwent trachelectomy with tumor size 2–4 cm had an almost six times greater risk of presenting a recurrence than those with tumor < 2 cm, even with a higher rate of adjuvant treatment. This adverse result observed in our series is independent from histology, lymphovascular space invasion, previous conization or surgical approach. Publications focusing on the oncological outcomes in the case of tumors larger than 2 cm are heterogeneous. The reported recurrence rates for patients with tumors larger than 2 cm vary tremendously, ranging from 0% to 38% [16]. Due to risk factors inherent to size, the literature often reports few patients who undergo surgery alone, without adjuvant treatment added [17].

When analyzing progression-free survival regarding tumor size in three categories (less than 20 mm vs. 20–29 mm vs. ≥ 30 mm), the 2 cm border remained significant. In the new FIGO classification 2018 [18,19], a new subdivision for stage IB disease every 2 cm increments in tumor size have been included: stage IB1 (≤ 2 cm), stage IB2 disease (> 2 cm to ≤ 4 cm) and stage IB3 (≥ 4 cm). This is due to the fact that tumors 2–4 cm present a greater association with other risk factors—such as LVSI (40% in our study) and parametrial or nodal involvement—that may require adjuvant therapy if they undergo primary resection, and, therefore, have different characteristics and outcomes from those smaller than 2 cm [20,21]. Therefore, and as shown in our results (statistically significant differences at 3 years PFS between groups (≤ 2 cm vs. 2–4 cm), the trend of the new guidelines is to advise against primary FSS.

Neoadjuvant chemotherapy followed by FSS has been explored in the last years as an option for patients with larger tumors who do not meet the criteria to perform conservative surgery alone. The role of neoadjuvant chemotherapy before fertility-sparing surgery in early cervical cancer patients is of great interest and deserves further investigation. In the ongoing CONTESSA study (NCT 04483557), the effectiveness of neoadjuvant chemotherapy in reducing the size of the tumor is explored. Patients with 2–4 cm tumor size will undergo three cycles of neo-adjuvant chemotherapy, and those presenting a complete or partial response (residual lesion < 2 cm) will receive fertility-sparing surgery. Results will be mature in 2025 [22].

In our study, the more frequent surgical approach reported was the vaginal route, followed by the laparoscopic route. We did not find any differences regarding surgical approach. Since Dargent described the vaginal radical trachelectomy, the surgical approach in fertility-sparing surgery in early cervical cancer has been a controversial issue. Some studies have postulated that an abdominal route improves oncological outcomes in patients with tumors bigger than 2 cm, due to the greater radicality of parametrial and paracervical resection [17]. In a recent review, it is postulated that selected patients with IB1 tumor size larger than 2 cm, with the absence of other risk factors such as deep one-third stromal invasion and lymphovascular space invasion, may be considered for an abdominal radical trachelectomy [23]. Aware of the publication of the LACC trial and the suggested negative effect of the uterine manipulator on cervical cancer recurrence, the results obtained in the FSS series cannot be compared [24]. It is not yet known whether true differences exist based on the route of radical trachelectomy; however, the ongoing International Radical Trachelectomy Assessment study will compare disease-free survival among patients with FIGO (2009) stage IA2 or IB1 (≤ 2 cm) cervical cancer who underwent open versus minimally invasive radical trachelectomy as primary endpoint, and will provide evidence in this field [25].

Obstetrical outcomes do not seem to be influenced by tumor size in terms of pregnancy rate, abortions or term of delivery. Obstetrical outcomes after radical trachelectomy range broadly in the literature and often do not reflect true pregnancy rates; this is because it may be influenced by the proportion of patients attempting to conceive, time of follow-up and lack of information on previous fertility [23].

This study shows real-world data of a large number of primary FSS performed in patients with early cervical cancer after negative lymph node assessment. Nevertheless, it is a retrospective study including patients treated with different protocols; therefore, its conclusions need to be consolidated by additional future studies.

## 5. Conclusions

In conclusion, in countries such as Spain, where childbearing is increasingly being delayed, fertility-sparing surgery is becoming a cornerstone of treatment in early cervical cancer patients. Fertility-sparing surgery should allow patients to both survive their cancer and preserve their uterus for future childbearing; however, selection criteria should be rigorous, especially for those patients with a tumor larger than 2 cm, due to the worse oncological outcomes associated with this. Upfront FSS should be discouraged in those patients with 2–4 cm tumor size. Other options may be discussed with the patient to reach a balance between the risk of recurrence and the best fertility results.

## Figures and Tables

**Figure 1 cancers-14-02108-f001:**
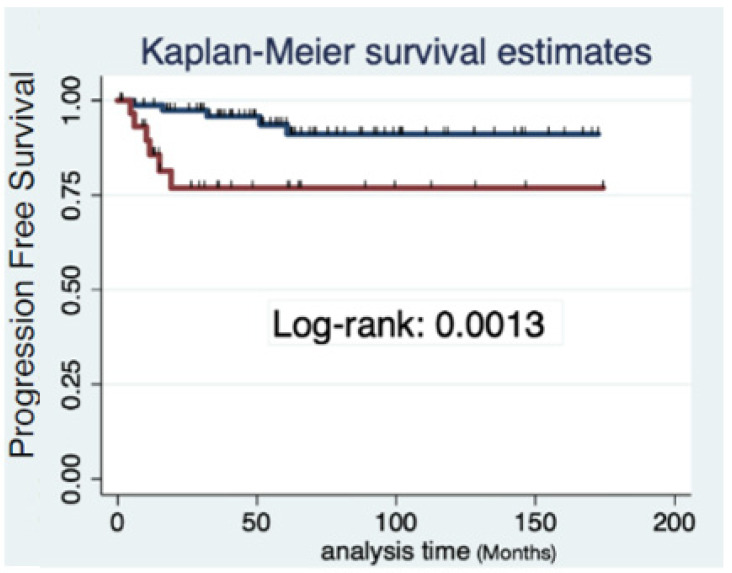
Progression-free survival in both groups. Kaplan–Meier estimates. log Rank: 0.011.

**Figure 2 cancers-14-02108-f002:**
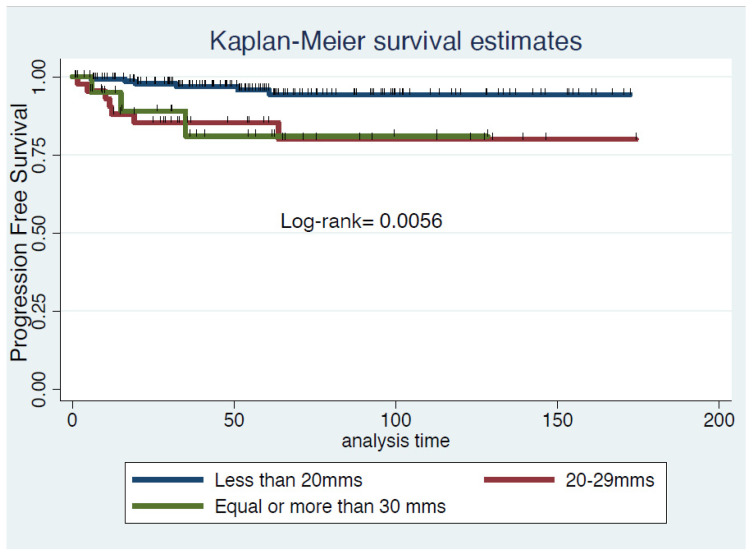
Progression-free survival in three groups. Kaplan–Meier estimates. log Rank: 0.006.

**Table 1 cancers-14-02108-t001:** Basal and surgical characteristics of both populations. LVSI: Lymph Vascular Space Invasion. MRI: Magnetic Resonance Imaging.

Trachelectomy	Tumor Size < 2 cm	Tumor Size 2–4 cm	*p*-Value
Number of Patients (Percentage)	82 (73.9)	29 (26.1)	
Age at diagnosis, years, mean (SD)	33.7 (5.3)	30.0 (4.8)	***p* = 0.001**
Histology, *n* (%):			
Squamous carcinoma	50 (61)	16 (55.2)	*p* = 0.662
Adenocarcinoma	32 (39)	13 (44.8)	
LVSI (+), *n* (%)	6 (8.2)	10 (40)	***p* = 0.001**
Previous conization, *n* (%)	39 (73.6)	13 (61.9)	*p* = 0.322
MRI, *n* (%)	78 (95.1)	28 (96.6)	*p* = 0.750
Neoadjuvant chemotherapy, *n* (%)	0	1 (3.5)	*p* = 0.261
Node assessment, *n* (%)			
Pelvic lymphadenectomy	59 (73.8)	21 (72.4)	*p* = 0.889
Only Sentinel Node Biopsy	21 (26.3)	8 (27.6)	
Trachelectomy, *n* (%)			
Simple	11 (13.4)	0	***p* = 0.030**
Radical	71 (86.6)	29 (100)	
Surgical approach, *n* (%)			
Laparoscopic	21 (25.6)	12 (41.4)	*p* = 0.281
Robotic	5 (6.1)	1 (3.5)	
Vaginal	56 (68.3)	16 (55.2)	
Cerclaje, *n* (%)	68 (82.9)	23 (79.3)	*p* = 0.663
Intraop. complications, *n* (%)			
Bladder injury	3 (3.7)	3 (10.3)	*p* = 0.399
Uterine artery injury	2	2	
Anaphylactic shock	1	1	
Postop. complications, *n*(%)	0 (0)	0(0)	*p* = 1
Adjuvant therapy	2 (2.4)	7 (24.1)	***p* < 0.001**

**Table 2 cancers-14-02108-t002:** Recurrence locations of both populations.

Recurrence Location, *n* (%)	Tumor Size < 2 cm *n* 82	Tumor Size 2–4 cm *n* 29	*p*-Value
Local	4 (80%)	5 (83.3%)	
- Cervical	2 (40%)	1 (16.7%)	
- Adnexal	1 (20%)	1 (16.7%)	
- Vaginal or paracervical	1 (20%)	3 (50%)	
Nodal	1 (20%)	1 (16.7%)	
Distant	0 (0%)	0 (0%)	
TOTAL	5	6	*p* = 0.84

**Table 3 cancers-14-02108-t003:** Simple Cox regression for predictors. LVSI (Lymph Vascular Space Invasion).

Variables	Hazard Ratio (CI 95%)	Hazard Ratio Adjusted (CI 95%)	*p*-Value
Tumoral size 2–4 cm	4.16 (1.26–13.69)	**5.99 (CI 95% 1.01–35.41)**	***p* = 0.02** ***p* = 0.036**
Adenocarcinoma histology	1.99 (0.61–6.56)	-	*p* = 0.25
LVSI positive	2.17 (0.56–8.42)	-	*p* = 0.26
Previous conization	0.34 (0.08–1.53)	-	*p* = 0.16
Simple vs. radical trachelectomy	0.76 (0.1–5.96)	-	*p* = 0.79
Surgical approach: Laparoscopy	ref		
Robotic approach	1.18 (0.14–10.09)		*p* = 0.88
Vaginal approach	0.40 (0.11–1.37)	-	*p* = 0.14

## Data Availability

Not applicable.

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
