# Peer review of "Tumor Size and Oncological Outcomes in Patients with Early Cervical Cancer Treated by Fertility Preservation Surgery: A Multicenter Retrospective Cohort Study"

_cancers, 2022, doi:10.3390/cancers14092108_

Round 1

Reviewer 1 Report

This study by Gil-Ibañez et al. discusses the impact of tumor size on the outcomes of fertility sparing surgery for CaCx patients. The main result is that patients with tumors >2cm diameter fare give significantly worse using FSS compared to patients with tumors <2cm.

The study is well designed, well presented and the discussion is adequate. I only have a few minor comments for correction and recommend this work for publication.

  • Line 28 abbreviation is used before term in line 30
  • Line 46: no comma after approximately and caners
  • Figure 1 and 2: graph formatting should be uniform. Use same font size, style, censored tick marks etc. Graph text also looks a little blurry.

Author Response

Dear Reviewer,

thank you for your comments on our study.

All your suggestions have been changed,

please find attached the reviewed manuscript,

Best regards,

Blanca Gil-Ibañez

Reviewer 2 Report

The authors are commended for there work.  This is an issue that the authors noted is becoming of greater import.  They did an excellent review of literature and implications to their own study.  The fact that this is a retrospective review is noted.  They also Discussed the LACC study which is important as the majority of patients who underwent definitive therapy with a laparoscopic approach.  They also noted the current NCCN guidelines.  While NCCN notes that several studies have shown that lesions from 2-4 have been done,   Outcomes were not statistically different between the 2 cohorts.  I would ask if the authors had any gastrointestinal or small cell tumors neuroendocrine type as NCCN specifically addresses these as non candidates for FSS.  Again I feel that this is a well done report and further contributes to our knowledge and management of these patients.

Author Response

Dear reviewer,

thank you for your comments on our study.

We just included histologically confirmed squamous carcinoma, adenocarcinoma or adenosquamus carcinoma in our cohort to be more uniform and in consensus with the international guidelines.

Best regards,

Blanca Gil-Ibañez

Reviewer 3 Report

Gil-Ibañez et al. have studied surgical, oncological and obstetrical outcomes of fertility-sparing surgery (FSS) in early cervical cancer in Spain. The tumor size was the most important negative prognostic factor in fertility-sparing surgery in cervical cancer. Patients with early cervical cancer tumor size >2 cm should be advised against primary FSS.

The claims are properly placed in the context of the previous literature. The experimental data support the claims. The manuscript is written clearly enough that most of it is understandable to non-specialists. The authors have provided adequate proof for their claims, without overselling them. The authors have treated the previous literature fairly. The paper offers enough details of methodology so that the experiments could be reproduced.

Comments

This paper would benefit from review by a native English speaker to assist with spelling and sentence construction. I am not. You have to choose between British English and American English. Now the manuscript is a mixture.

Minor revisions

Line 27-28, "Patients with early cervical cancer tumour size >2cm and gestational desire should be discouraged about primary FSS" => "Patients with early cervical cancer tumor size >2 cm and desire for pregnancy should be advised against primary FSS"

Line 46, "Approximately, 15% of all cervical cancer occur in women under 40 years of age"

This may be the case globally, but in western countries with organized screening (including Spain), the incidence in women older than 40 years of age have been reduced. However, screening has little impact of cervical cancer incidence in women younger than 40 years of age, and the proportion of cervical cancer in women under 40 years of age is therefore higher.

In Norway, in 2016-2020, 35% (642/1799) of all cases of cervical cancer occur in women under 40 years of age. In Spain I think the proportion of cervical cancer in women under 40 years of age is closer to 35% than 15%.

Line 85, missing space "stage IA1 with lymphovascular space invasion, IA2 and IB1 (< 2cm and 2-4 cm)" => "stage IA1 with lymphovascular space invasion, IA2 and IB1 (< 2 cm and 2-4 cm)"

Line 142, "patients with tumor 2-4cms (33.7 vs. 30.0 years, p<0.05)" => "patients with tumor 2-4 cm (33.7 vs. 30.0 years, p<0.05)"

Line 149, Table 1, "Neoadjuvant quemotherapy" => "Neoadjuvant chemotherapy"

Line 152, "In one patient with a tumor 2-4cm" => "In one patient with a tumor 2-4 cm"

Line 157, "treatment in the group of tumors 2-4cm because of risk factors" => "treatment in the group of tumors 2-4 cm because of risk factors"

Line 160, "the last follow up (9.9%), 5 (6.0%) in the <2cms tumor group and 6 (21.4%) in the 2-4cms" => "the last follow up (9.9%), 5 (6.0%) in the <2 cm tumor group and 6 (21.4%) in the 2-4 cm"

Line 175-176, "No differences were observed in terms of site of recurrence between tumors <2cm and tumours 2-4 cm" => "No differences were observed in terms of site of recurrence between tumors <2 cm and tumors 2-4 cm"

Line 179, Table 2, "Tumor size < 2cms" => "Tumor size < 2 cm"

Line 179, Table 2, "Tumor size 2-4cms" => "Tumor size 2-4 cm"

Line 182, "Only tumor size (<2cm vs. 2-4cms) was found to be significant" => "Only tumor size (<2 cm vs. 2-4 cm) was found to be significant"

Line 184, "rest of the variables, tumor size ≥2cm has a Hazard Ratio" => "rest of the variables, tumor size ≥2 cm has a Hazard Ratio"

Line 186, Table 3, "Simple vs radical traquelectomy .76 (0.1-5.96)" => "Simple vs radical trachelectomy 0.76 (0.10-5.96)"

Line 186, Table 3, "Vaginal approach .40 (0.11-1.37)" => "Vaginal approach 0.40 (0.11-1.37)"

Line 189, "case occurred in a 25-year-old woman with a stage IB1 ≥2cm (30 mm)" => "case occurred in a 25-year-old woman with a stage IB1 ≥2 cm (30 mm)"

Line 200, "Thirty-one pregnancies were recorded in the <2cms group" => "Thirty-one pregnancies were recorded in the <2 cm group"

Line 201, "the 2-4cms group" => "the 2-4 cm group"

Line 203, "of 13 preterm births (35.5% vs 15.3%) and 12 full-term births (25.85 vs 30.8%)" => "of 13 preterm births (35.5% vs 15.3%) and 12 full-term births (25.9% vs 30.8%)"

Line 206, "This study found that patients with early cervical cancer with tumor size ≥2cm" => "This study found that patients with early cervical cancer with tumor size ≥2 cm"

Line 225, "rates for patients with tumors bigger than 2cm vary tremendously" => "rates for patients with tumors bigger than 2 cm vary tremendously"

Line 262, "survival among patients with FIGO (2009) stage IA2 or IB1 (≤2cm) cervical cancer" => "survival among patients with FIGO (2009) stage IA2 or IB1 (≤2 cm) cervical cancer"

Author Response

Dear reviewer,

thank you for your comments on our study.

The paper has been reviewed by a native English speaker to assist us with spelling and sentence construction. Thank you for pointing it out. 

Line 27-28: changed

Line 46: as you pointed, it refers to the globbal incidence. We looked for the exact data in Spain but we couldn´t get them so we thought it was more appropiate to adress the problem with a general point of view. 

Line 85,142,149,152, 157, 160, 175, 179, 182,84,186,189,200,201,203.206,225 and 262 : done, thank you

Thank you for your help,

best regards,

Blanca Gil-Ibañez